# CLEANCoNLL: A Nearly Noise-Free Named Entity Recognition Dataset

**Susanna Rücker**
Humboldt-Universität zu Berlin
susanna.ruecker@hu-berlin.de

**Alan Akbik**
Humboldt-Universität zu Berlin
alan.akbik@hu-berlin.de

## Abstract

The CoNLL-03 corpus is arguably the most well-known and utilized benchmark dataset for named entity recognition (NER). However, prior works found significant numbers of annotation errors, incompleteness, and inconsistencies in the data. This poses challenges to objectively comparing NER approaches and analyzing their errors, as current state-of-the-art models achieve F1-scores that are comparable to or even exceed the estimated noise level in CoNLL-03. To address this issue, we present a comprehensive relabeling effort assisted by automatic consistency checking that corrects 7.0% of all labels in the English CoNLL-03. Our effort adds a layer of entity linking annotation both for better explainability of NER labels and as additional safeguard of annotation quality. Our experimental evaluation finds not only that state-of-the-art approaches reach significantly higher F1-scores (97.1%) on our data, but crucially that the share of correct predictions falsely counted as errors due to annotation noise drops from 47% to 6%. This indicates that our resource is well suited to analyze the remaining errors made by state-of-the-art models, and that the theoretical upper bound even on high resource, coarse-grained NER is not yet reached. To facilitate such analysis, we make CLEANCoNLL publicly available to the research community[1].

## 1 Introduction

The CoNLL-03 corpus is arguably the most well-known and utilized dataset for named entity recognition (NER). CoNLL-03 is typically the crucial benchmark to evaluate new NER approaches (Schweter and Akbik, 2020; Zhou and Chen, 2021; Wang et al., 2021), language models (Conneau et al., 2020; He et al., 2023), and – most recently – instruction-tuned LLMs for their few- and zero-shot abilities (Qin et al., 2023; Ashok and Lipton, 2023).

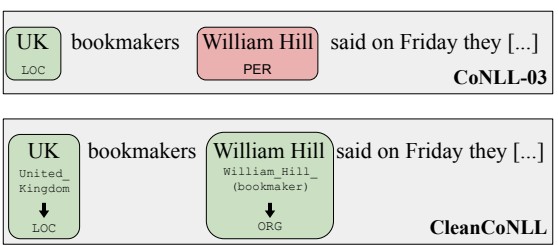

Figure 1: An example from CoNLL-03 in which the named entity "William Hill" is mislabeled as a person (PER). In CLEANCoNLL, we leverage the entity link to the Wikipedia article of the company to determine that this entity is indeed of type organization (ORG). Further, our entity linking annotation makes more immediately transparent to human inspection why this entity is labeled as ORG.

However, prior works have found significant data quality and consistency issues: Reiss et al. (2020) identified a share of 3.8% of all labels as incorrect, with particularly high error rate in the test split (7.5% error rate). Similarly, Wang et al. (2019) found errors in 5.4% of all sentences of the test split. Qualitatively, various prior works noted inconsistencies where different labels are given to the same entity throughout the corpus, affecting for instance the names of sports teams when referred to by their geographical name (Wang et al., 2019), sports leagues (Wang et al., 2019), names of deities and religious contexts (Ratinov and Roth, 2009), and the names of ministries (Stanislawek et al., 2019). Further, various prior works noted segmentation errors in which token-, entity- and even sentence-boundaries are incorrectly set (Stanislawek et al., 2019; Reiss et al., 2020).

We argue that these inconsistencies impair the usefulness of CoNLL-03 for benchmarking state-of-the-art (SOTA) NER approaches, since such approaches generally report F1-scores that lie near (or even above) its estimated noise level. For instance, we find in this paper that nearly half of errors made by a state-of-the-art NER model are in fact correct

---

[1] https://github.com/flairNLP/CleanCoNLL

predictions that are falsely counted as errors due to noisy annotation. We argue that given the maturity of research into high-resource NER, the research community requires a dataset with highly consistent, nearly noise-free labels in order to correctly estimate the performance of state-of-the-art models and to better understand their remaining errors.

**Adding and leveraging entity links.** To address this issue, we present CLEANCoNLL, a revised version of CoNLL-03 that significantly improves annotation quality and consistency. Our main idea is to introduce a second layer of annotation to the corpus that adds entity linking information. Entity linking is the task of disambiguating named entities to a unique ID in a knowledge base, typically the Wikipedia page representing a specific entity.

As Figure 1 illustrates with an example sentence, adding entity links serves two purposes: First, entity links allow us to define automatic consistency checks on NER labels, since prior work found that NER labels can be (partially) derived from Wikipedia page IDs (Tedeschi et al., 2021). Second, we find that the disambiguating information provided by entity links makes NER labels more immediately transparent to human inspection, thus making it easier for annotators to spot mistakes and discuss disagreements.

**Contributions.** In more detail, our contributions are the following:

- We present CLEANCoNLL, the result of a comprehensive relabeling effort of CoNLL-03 that includes updated, more consistent NER annotations in all three splits and integrates an additional layer of entity linking annotation.

- We detail a method for deriving NER annotation from Wikipedia links that leverages the category system in Wikidata, and discuss how this approach acts as consistency checks to safeguard annotation quality.

- We further describe an automatic approach to flag potential inconsistencies that we use throughout the manual re-annotation process, which we refer to as "cross-checking".

- We conduct extensive analyses of CLEAN-CoNLL and present both quantitative results of manual assessment of annotation quality in comparison to previous versions and qualitative examples of error corrections. We train a set of current state-of-the-art approaches on

CLEANCoNLL, and comparatively evaluate the true error rate of these approaches.

We make the resource publicly available and encourage the community to use and further improve its quality.

## 2 Creating CLEANCoNLL

CLEANCoNLL was created in three distinct phases of annotation, which we describe chronologically in the following subsections:

1. We first (see Sec. 2.1) added a layer of entity linking annotation by merging two existing resources, and developed a heuristic to automatically correct NER errors using entity links.

2. We then (see Sec. 2.2) performed 3 iterative rounds of cross-checking to identify potential inconsistencies, which we manually examined and corrected whenever necessary[2].

3. Finally (see Sec. 2.3), we developed special handling for adjectival affiliation entity names.

### 2.1 Phase 1: Adding Entity Links

**Base corpus.** We use the version from Reiss et al. (2020) as the starting point of our relabeling effort[3]. Reiss et al. (2020) already corrected 3.8% of all labels in CoNLL-03, including several major sentence and entity mention boundary problems. While this version has less noise than the original corpus, our own analysis (Sec. 3.4) and prior work (Wang and Mueller, 2022) found that large numbers of errors still remain.

**Merging AIDA.** For initial corpus creation, we take advantage of the fact that AIDA (Hoffart et al., 2011), a well-known entity linking dataset, shares the same text base as CoNLL-03. Since the entity links in AIDA are expert-annotated by a group distinct of the original CoNLL-03 annotators, they provide a layer of high quality annotation of an arguably more complex task.

However, merging these two corpora was not straightforward since there are discrepancies both in terms of sentence boundary definitions as well

---

[2]Manual annotation in all phases of the process was carried out by the authors of this work. See the limitations section for a brief discussion.

[3]Obtained by using their instructions from `https://github.com/CODAIT/Identifying-Incorrect-Labels-In-CoNLL-2003`

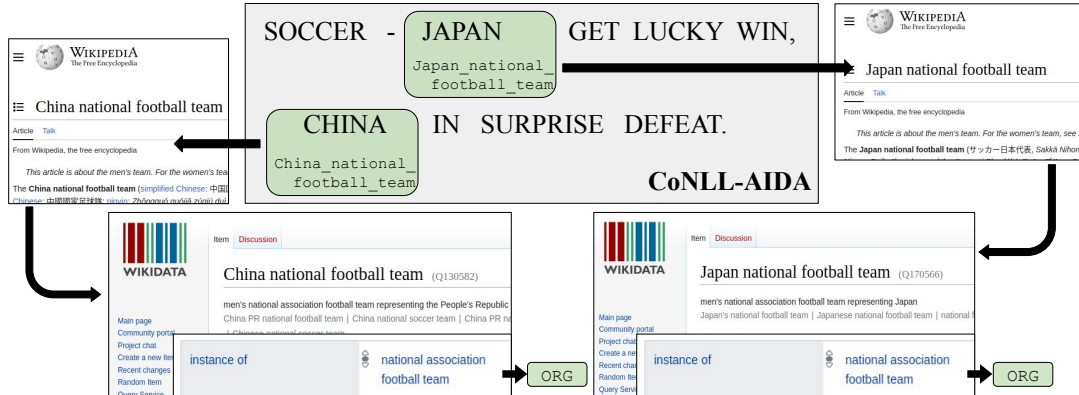

Figure 2: A good portion of named entities in CoNLL-03 have manually assigned Wikipedia URLs in AIDA. In the example sentence "JAPAN" and "CHINA" are linked to the article of the referred sports teams from which we derive their Wikidata items and categories.

as the coverage of entity annotations. We found that of all 34,941 entities in the Reiss CoNLL version, 78.9% have an entity link in AIDA, while the remaining 21.1% (7,369 entities) do not. We found three main reasons for lack of coverage: *(1)* Rare entities that at time of labeling had no corresponding Wikipedia article are unlabeled in AIDA. *(2)* The definition of what constitutes a named entity differs in some instances, with CoNLL-03 for instance labeling the span "Boeing 707" as entity, while AIDA links only "Boeing". *(3)* Some common entities like "EU" for unclear reasons are consistently unlabeled in AIDA.

**Deriving NER labels from entity links.** We derive NER labels from Wikipedia URL links by leveraging Wikidata[4], a manually curated knowledge graph that connects entities (represented by Wikipedia pages) through a set of relations. We leveraged two of these relations: (1) `instance_of`, which connects an instance to the conceptual class it instantiates, (2) `subclass_of`, which connects a conceptual class to a more general conceptual class.

Refer to Figure 2 for an example of this: The mention "CHINA" is linked to the entity "China_national_football_team", which is a Wikidata instance of "national association football team", which in turn is a subclass of "national sports team". With these two relations, we automatically select conceptual classes for each linked entity in our corpus.

For the most common of these classes, we manually defined a mapping to CoNLL-03 entity types. For instance, we manually defined that the Wikidata category "sports team" maps to the entity type

ORG. A full description of our mappings is provided in Appendix A.1. Using these mappings, we automatically mapped 4,163 (74.4%) of the 5,595 unique Wikipedia labels in AIDA to their NER type. However, as we prioritized precision over recall in the development of our manual mappings, 1,432 (25.6%) could not be classified automatically because either none or multiple of our heuristics took effect.

**Manual annotation.** These remaining 1,432 entity links were manually assigned NER types. This annotation was carried out separately for all labels by two experts who were provided with guidelines as well as the respective Wikipedia and Wikidata information. The inter-annotator agreement was 94.9%, cases of disagreement were subsequently resolved in discussion.

Using the resulting mappings, we updated all NER labels in the corpus to be consistent with the entity link.

## 2.2 Phase 2: Consistency Checking

After Phase 1 of our relabeling effort, a number of potential error sources remained:

- AIDA entity links, from which we derive NER labels in Phase 1, were missing for a subset of entities, and may themselves contain noise. For instance, we qualitatively observed instances in which national sports teams were incorrectly linked to the Wikipedia article of the country instead of the team.

- Though developed to favor precision, our heuristics of deriving NER labels from Wikidata categories may lead to an incorrect NER label assignment.

---

[4] https://www.wikidata.org/

- Leveraging AIDA entities is ill-suited to correct wrongly spanned mention boundaries or tokenization problems.

### 2.2.1 Cross-Checking and Correction

To identify and correct these errors, we follow a "cross-checking" approach as introduced in Wang et al. (2019). The main idea is to predict labels for a small part of the corpus using the entirety of the remaining corpus, but make sure that no entity names are memorized.

To accomplish this, we collect all unique surface forms $\mathcal{S}$ of mentions in the corpus and group them randomly into 20 distinct batches $\mathcal{S}_i$ of around 500 surface forms each. For each of these 20 batches of surface forms, we create two splits: $\mathcal{D}_i^{test}$, which contains all sentences that have one of the surface forms $\mathcal{S}_i$ of the respective batch. $\mathcal{D}_i^{train}$, which contains all other sentences, i.e. the sentences that contain only $\mathcal{S} \setminus \mathcal{S}_i$ (in other words: do not contain any of the surface forms of the respective batch). For each batch, we train a state-of-the-art model on $\mathcal{D}_i^{train}$ and use it to predict on $\mathcal{D}_i^{test}$. All predictions that differ from the annotated NER labels are flagged as potential errors.

**Manual error resolution.** All annotations flagged as potential errors were then manually inspected by annotators. They were given the entity span within document context and the different label possibilities including their source: the current gold annotation, the diverging model prediction, as well as the Wikipedia label and its automatically derived NER label. Based on this information, they were asked to a) decide which NER label to use, and b) add or change the current Wikipedia label if necessary. See Figure 3 for an illustration of this interface.

**Annotation process.** In the first round of annotation, all spans were independently labeled by two annotators (agreement was 79.3%), after which the guidelines were refined and the rest was done primarily by one. Annotators were given the option to mark cases of doubt (2.4%); these were given to the second annotator for independent annotation.

Annotators were further asked to take note of error classes that may go beyond a single instance. For instance, we noted general problems with annotation of time zones ("GMT", "EST"): Some were labeled as entities and others were not. In such cases, we clarified the annotation guidelines and manually corrected all instances in the corpus.

```
You have annotated 75 out of 1020 spans!
###################

span:  England      labels:     gold: LOC,
                                pred: ORG,
                                aida: England
                                https://en.wikipedia.org/wiki/England

(nr 2 of 4 with this mention and pattern)
###################

In context and marked:
    England manager Glenn Hoddle called up uncapped Everton
    defender Andy Hinchcliffe on Sunday to the national squad for
    the opening World Cup qualifier against Moldova next weekend .

[?] What to do with this span?: use  Pred: ORG
    keep Gold: LOC
 >  use  Pred: ORG
    exactly like last: label: LOC, aida: England
    use MISC
    use PER
    no label ("O")
    add/change url (NER can be changed afterwards)
    show more context
    show (more) URLs nearby
    show candidate URLs
    mark: SECOND OPINION (still needs your annotation!)
    <<< previous
    clear screen
```

Figure 3: Example of the annotator interface during cross-checking. The mention "England" is annotated as "LOC", but the model predicted "ORG", which is correct. The wrong annotation originates from an error in AIDA, where the mention is linked to the country's article ("England") instead of the football team. Our annotator decided on NER label "ORG" and corrected the entity link to the Wikipedia page "England_national_football_team".

### 2.2.2 Iterative Cross-Checking in 3 Rounds

We executed 3 rounds of cross-checking. After each round, labels were manually corrected and the improved corpus used to train the next models:

**1. Entity detection and classification.** The first round of cross-checking directly followed Wang et al. (2019) to train a model to jointly detect mention boundaries as well as classify them using a BIOES sequence tagging approach (Schweter and Akbik, 2020). Over all batches, this resulted in 1,020 flagged annotations, resulting in 521 corrected labels after manual inspection.

**2. Only entity classification.** The second round of cross-checking instead trained a model to only classify already provided entity mentions. See Appendix A.7 for details of this model. This flagged a further 709 instances for inspection, resulting in 98 additional corrected NER labels.

**3. Only entity detection.** The third round of cross-checking focused only on consistent definitions of entity mentions. To this end, we trained a BIOES sequence labeler to predict only entity boundaries (but not their type). Over all batches, we collected 726 sentences with need of manual decision concerning mention boundaries (1,720 spans when counting the predicted and gold ones separately

| | Sentence (part) | Mentions | Entity Link (Wikipedia URL) | NER before Phase 3 | NER after Phase 3 |
|---|---|---|---|---|---|
| a) | [...] while **Italian** team mate Roberto Baggio did not play due to injury. | [**Italian**] [Roberto Baggio] | Italy_national_football_team Roberto_Baggio | **ORG** PER | **MISC** PER |
| b) | Dick Morris, the **Republican** political consultant who reshaped [...] | [Dick Morris] [**Republican**] | Dick_Morris Republican_Party_(United_States) | PER **ORG** | PER **MISC** |
| c) | Rosati will meet **Serbian** President Slobodan Milosevic | [Rosati] [**Serbian**] [Slobodan Milosevic] | Dariusz_Rosati Serbia Slobodan_Milošević | PER **LOC** PER | PER **MISC** PER |

Table 1: Example sentences to illustrate the differences in adjectival affiliation entities. In the original CoNLL-03, affiliations such as "Italian", "Republican" and "Serbian" are marked MISC. However, since we derive NER tags from entity links, they are initially marked as ORG ("Italian" here for instance refers to the national football team of Italy) or LOC. Phase 3 of our relabeling reverts these back to MISC usage.

despite their overlaps) that were resolved by manual annotation. In 450 of these sentences, different boundaries of one (or more) mentions were set.

### 2.3 Phase 3: Handling Adjectival Affiliations

After Phase 2 of our relabeling effort, a major inconsistency to the original CoNLL-03 definitions became apparent: Adjectival affiliation entities are consistently labeled MISC in CoNLL-03, whereas our corpus assigns the NER tags of their nominal usage. As Table 1 illustrates, adjectival entities such as "Italian" may be labeled ORG in our corpus if linked to the Wikipedia page of the national football team of Italy.

To make our annotations consistent with original CoNLL-03 definitions, we semi-automatically revert such cases to the MISC label. Using two prediction models trained on OntoNotes (Hovy et al., 2006), we predict adjectives and entities of OntoNotes type "NORP" (affiliations to "nationalities or religious or political groups") to flag entities that should potentially be reverted to MISC. We automatically reverted flagged entities if they were originally labeled MISC in the Reiss version and manually checked all others.

## 3 Evaluation

We aim to get an understanding of annotation quality of CLEANCONLL. To this end, we discuss its properties (Sec. 3.1) and perform a manual evaluation of 100 sentences to assess and compare annotation quality with earlier variants (Sec. 3.2-3.4).

### 3.1 Dataset Statistics

**Label distribution.** Table 2 compares the total count of annotated named entities and the distribu-

| | CoNLL-03 | | Reiss | | CLEANCONLL | |
|---|---|---|---|---|---|---|
| | abs | % | abs | % | abs | % |
| LOC | 10,645 | 30.3 | 10,103 | 28.9 | 9,399 | 26.7 |
| ORG | 9,323 | 26.6 | 9,922 | 28.4 | 10,492 | 29.8 |
| PER | 10,059 | 28.7 | 9,983 | 28.6 | 9,947 | 28.2 |
| MISC | 5,062 | 14.4 | 4,933 | 14.1 | 5,419 | 15.4 |
| # entities | 35,089 | | 34,941 | | 35,257 | |
| # sent. | 20,744 | | 20,617 | | 20,617 | |

Table 2: Statistics of CLEANCONLL, in comparison to CoNLL-03 and the Reiss version.

tion across the four classes for CLEANCONLL, the original CoNLL-03 and the Reiss variant.

We observe that CLEANCONLL has a slightly higher number of ORG and MISC entities than the base versions. Among other reasons, this stems from a more consistent use of ORG labels for sports teams referred to by their geographic name, and more consistent use of MISC for entity types such as named sports leagues and stocks. The appendix provides more information on such cases in Appendix A.6, and discusses qualitative examples in Appendix A.5.

**Updated NER labels.** Table 3 further examines the extent of label updates introduced by our approach. Compared to CoNLL-03, the labels of 1,832 entity mentions (5.2%) were updated, and 625 (1.8%) mentions were newly added[5]. In sum, we thus modified 7% of labels, affecting 10.3% of sentences[6]. We specifically note slightly higher label update rates in the test split (9.4% modified labels, affecting 14.2% of sentences) and the dev

---

[5]This includes both completely new ones as well as mentions where boundaries were moved.

[6]Refer to Appendix A.8.3 for the number of changed labels *before* reverting MISC to its original definition in Phase 3 (14.9%).

| CLEANCONLL vs. | CoNLL-03 | | Reiss | |
|---|---|---|---|---|
| | abs | % | abs | % |
| Label unchanged | 32,110 | 91.5 | 33,488 | 95.8 |
| Label updated | 1,832 | 5.2 | 1,205 | 3.4 |
| New mention | 625 | 1.8 | 534 | 1.5 |
| Unknown | 690 | 2.0 | 30 | 0.1 |
| # updated sent. | 2,115 | 10.3 | 1,398 | 6.8 |

Table 3: NER label updates, comparing CLEAN-CONLL with CoNLL-03 and Reiss. *Unknown*: Due to updated sentence segmentation, for some entities we cannot determine a direct equivalent, so we exclude these mentions here. *New mention*: Includes both completely new and newly bounded mentions.

split (8.4% modified labels, affecting 12.5% of sentences). Compared to Reiss, we changed 1,205 labels (3.4%) and added 534 mentions (1.5%).

**Updated entity links.** Our annotation process added 1,955 new entity links over the AIDA base version and corrected 322 links. See Table 10 in Appendix A.8.2 for more details.

### 3.2 Evaluation of Annotation Quality

We estimate the annotation quality for four versions of the corpus: The base CoNLL-03 version, the Reiss variant, CLEANCONLL, and a variant of our corpus without Phase 3 of annotation, i.e. without reverting adjectival affiliation names to MISC. We refer to this variant as CLEANCONLL*.

**Manual evaluation.** We gather all sentences in which there is at least one difference in annotation between at least two of the corpora. From these, we sampled 100 sentences for manual evaluation. For all 4 corpora, all 100 sentences were independently labeled by two expert annotators. Cases of disagreement were discussed and, if possible, resolved. In a subset of cases, annotators found contexts ambiguous, meaning that two different NER interpretations might be seen as valid. These cases were specifically marked.

**Results.** Figure 4 illustrates the results for all 4 variants of the corpus. We find about 10% of mentions marked as incorrect or missing in the original CoNLL-03, and about 7% of mentions either incorrect or missing in Reiss. Our CLEANCONLL version by contrast was found to have only about 1% annotation errors in both variants, and no missing mentions.

These findings indicate a significant improvement in annotation quality on CLEANCONLL compared to earlier CoNLL-03 versions.

### 3.3 Discussion of Ambiguities

Our manual evaluation also showed that cases of ambiguities remain in CLEANCONLL that may be debated in our annotation guidelines. For instance, we consistently label named airports as ORG, even in sentences such as "the flight to Atlanta" (where humans might interpret "Alanta" as LOC instead of ORG). We decided to label time zones ("GMT") as entities, but do not label physical units.

A particular problem we faced are different possible segmentations of larger entities. The span "Florida Supreme Court" may be seen as one entity, or may be seen to be composed of the LOC-entity "Florida" and the ORG-entity "Supreme Court". In such cases, we generally favored the longer entity. One avenue for future work might be to add nested entities so that both readings of this span could be reflected in the data.

### 3.4 Further Qualitative Discussion

We conduct a more extensive qualitative discussion of labeling decisions in CLEANCONLL, and how they differ from CoNLL-03, in Appendix A.5 and Table 7. There, we discuss examples that cover different types of annotation problems, like outright wrong labels, missing mentions, boundary problems as well as inconsistent use of labels on similar or even the same instances.

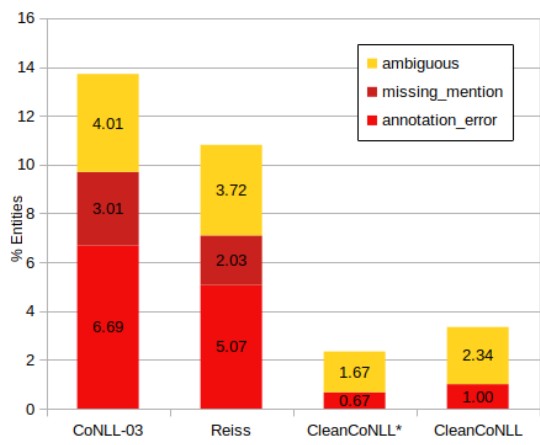

Figure 4: Estimation of annotation quality based on a manually evaluated sample of 100 sentences for each of the four CoNLL-03 variants. For visual clarity we exclude the correct spans. We find that CLEANCONLL has a significantly lower share of annotation errors than prior versions.

| Name, Citation | Base embeddings | CoNLL-03 | Wang | Reiss | CLEANCoNLL* | CLEANCoNLL |
|---|---|---|---|---|---|---|
| **Flair** (Akbik et al., 2018) | Flair Embeddings, GloVe embeddings | $92.77 \pm 0.12$ | $93.75 \pm 0.26$ | $92.92 \pm 0.1$ | $93.27 \pm 0.22$ | $\mathbf{94.21 \pm 0.28}$ |
| **FLERT** (Schweter and Akbik, 2020) | Transformer ("xlm-roberta-large") | $93.94 \pm 0.27$ | $95.28 \pm 0.27$ | $94.32 \pm 0.09$ | $96.78 \pm 0.07$ | $\mathbf{96.98 \pm 0.05}$ |
| **Biaffine** (insp. by Yu et al. (2020)) | Transformer ("xlm-roberta-large") | $93.84 \pm 0.18$ | $95.0 \pm 0.13$ | $94.34 \pm 0.17$ | $96.59 \pm 0.08$ | $\mathbf{97.08 \pm 0.02}$ |
| **ACE** (sentence) (Wang et al., 2021) | combination of several | $93.56 \pm 0.15$ | $94.77 \pm 0.06$ | $93.48 \pm 0.16$ | $95.41 \pm 0.25$ | $\mathbf{95.89 \pm 0.0}$ |

Table 4: Comparing standard NER models on the different corpus versions. Averaged F1-score and standard deviation from three runs (two runs for ACE).

## 4  Experiments

We train and evaluate a representative set of NER models on CLEANCoNLL, as well as on CoNLL-03 and the versions by Wang et al. (2019) and Reiss et al. (2020), to determine to what extent our relabeling effort affects model performance. We then sample a set of erroneous model predictions for each corpus variant to determine the share of true errors.

We select models from literature that report state-of-the-art F1-scores on CoNLL-03 and/or are easily reproducible with open source libraries:

(1) A **Flair** model as proposed by Akbik et al. (2018), using stacked bidirectional contextual string embeddings on character-level and GloVe embeddings (Pennington et al., 2014).

(2) A **FLERT** model as proposed by Schweter and Akbik (2020), using document-level context features for fine-tuning a transformer model.

(3) A **Biaffine tagging** approach, inspired by Yu et al. (2020), in which all possible entity spans up to a maximum length (5) are considered as potential entities and separately classified.

(4) The **ACE** model by Wang et al. (2021) that uses automatic selection and combination of a set of pre-trained embeddings[7].

### 4.1  Experimental Results

Table 4 reports averaged F1-scores and standard deviation over three random seeds for each experimental setting. As the table shows, we observe consistently higher F1-scores on CLEANCoNLL compared to all earlier variants of CoNLL-03. This gives indication that our relabeling effort successfully improved label and mention consistency.

### 4.2  Classification of Model Errors

To gain further insight, we sample 100 prediction errors made by a FLERT model trained on each of the four variants of CoNLL-03. We use FLERT since it achieves the best results on average over all CoNLL variants. Our goal is to determine whether errors are "true errors" rather than an artifact of noisy annotation in the test data.

**Manual evaluation.** All 400 errors were independently evaluated by two expert annotators. They were tasked to determine whether the error is in fact a real *model error* or rather an *incorrect annotation*. We added the possibility *both correct*, for cases in which both model and annotation can be argued for or there is ambiguity in the context, making a clear decision impossible[8].

**Results.** Figure 5 shows the result of our manual evaluation: The share of correct predictions incorrectly classified as errors drops significantly from 47% in the original CoNLL-03 to 31% in the Reiss variant, to only 6% and 7% in the CLEAN-CoNLL variants. This indicates a significantly higher annotation quality in CLEANCoNLL.

In addition, the increased share of true model errors indicates that CLEANCoNLL is well-suited to evaluate the remaining errors made by state-of-the-art models, as error classification is for the largest part reliable. The fact that most remaining errors are true errors instead of artifacts of noise indicates that the upper bound for coarse-grained NER is not yet reached and that higher F1-scores than the current state-of-the-art of 97% are theoretically feasible.

---

[7]We use their sentence-level setup, as their document-level setup was computationally too costly for us to reproduce for all variants and random seeds.

[8]For both model and annotation errors we distinguished *mention boundary* and *label* errors, but for visual simplicity aggregated them here under model or annotation error.

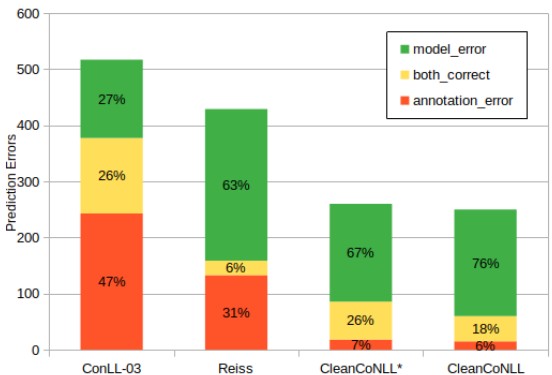

Figure 5: Estimated share of true vs. annotation errors in prediction errors for a model trained on each of the four CoNLL-03 variants. We find that the annotation error rate decreases from 47% in original CoNLL-03 to only 6%-7% for CLEANCoNLL, indicating significantly higher annotation quality.

## 5   Release of CLEANCoNLL

We make CLEANCoNLL publicly available to the research community through a github repository. We follow CoNLL-03 in releasing only annotations and a script to merge them with the Reuters corpus that can be obtained free of charge for research purposes[9] to produce the annotated corpus, see Figure 6 for an excerpt.

**Annotation layers**. We release three new layers of annotation: *(1)* Our entity linking annotations, *(2)* the final CLEANCoNLL NER tags, and *(3)* our NER tags after Phase 2, i.e. before reverting adjectival affiliations to MISC. We release the latter as we believe some researchers might find type-resolved adjectival affiliations useful, and further discuss advantages and disadvantages of both NER label versions in Appendix A.4.

**Continued improvements by the research community**. Taking inspiration from the community-driven process of improvements in the universal dependency treebanks project (Nivre et al., 2020), we set up our open source repository such that researchers may file pull requests to further improve annotation quality, and discuss ambiguous cases or underspecification in the annotation guidelines. Our hope is that active participation by researchers in NER and entity linking will over time further improve annotation quality and further reduce noise.

---

[9]For instructions on how to obtain, refer to https://www.clips.uantwerpen.be/conll2003/ner/

## 6   Related Work

As previously mentioned, other works have pointed out poor annotation quality in CoNLL-03 (Wang et al., 2019; Reiss et al., 2020; Wang and Mueller, 2022; Stanislawek et al., 2019). Most relevant for our work are Wang et al. (2019) and Reiss et al. (2020), who both published their own corrected versions of the corpus.

Wang et al. (2019) propose a method to improve training on noisy datasets via lowering the weights of potentially incorrect instances. They use their method to identify errors, but only focus on the test set. They release a manually corrected test set called CoNLL++. Similarly, Reiss et al. (2020) use majority voting in an ensemble of trained models to identify potential errors and release a manually corrected version of CoNLL-03.

Several other works also touch the topic of label error detection or model error classification in CoNLL-03 (Stanislawek et al., 2019; Fu et al., 2020; Jain et al., 2022; Chen et al., 2023). Liu and Ritter (2022) do not correct annotation errors but create a completely new dataset which they also call CoNLL++ (not to be confused with the corrections by Wang et al. (2019)), that emulated CoNLL-03 label usage and distribution as well as text domain, but consists of newer articles.

Our annotation effort differs in that we *(1)* first integrate a layer of entity linking information for consistency checking and to assist manual annotation, and *(2)* use trained models for cross-checking over multiple iterations during our manual re-annotation process. Our evaluation indicates that CLEANCoNLL has significantly higher annotation quality than all previous versions.

## 7   Conclusion

We presented CLEANCoNLL, a corrected version of the classic NER benchmark CoNLL-03, with updated and more consistent NER labels. Our approach adds an additional layer of entity linking annotation to disambiguate entities whenever possible to their Wikipedia page. We leverage this layer to both identify inconsistencies in annotation, and to make labeling decisions transparent to human inspection. Further, we apply an approach of cross-checking throughout our manual annotation process as additional consistency check.

Through our relabeling effort, we updated 7% of all labels of CoNLL-03, and added 5.5% new entity links over the base AIDA resource. Our eval-

uation strongly indicates that overall annotation quality and consistency is significantly improved in CLEANCONLL. In Sec. 3.2, we estimated a noise share of only around 1% for our sample, whereas CoNLL-03 had an estimated noise share of 10%. We found that current models reach significantly higher F1-scores on our dataset (Sec. 4.1), and that the share of correct predictions incorrectly classified as errors drops from 47% to 6% (Sec. 4.2).

These findings indicate the validity of our relabeling approach, and the usefulness of CLEAN-CONLL for training and evaluating state-of-the-art NER models. To enable the research community to do so, we make our new layers of annotation publicly available through an open source repository modeled after the UD treebanks. We hope that active participation by the research community will help us to continuously further improve annotation quality, with the ultimate goal of an entirely "noise-free" NER dataset.

## Limitations

Despite our best efforts to ensure consistency, it is possible that there still is inconsistency of label usage in CLEANCONLL, which we enumerate as limitations here. (1) It is possible, that specific instances are wrongly labeled (e.g. because of a wrong or missing Wikipedia label leading to an incorrect NER label that fell through our cross-checking approach). (2) Also – this might be inevitable – some of our annotation guideline decisions can be up for discussion[10], as well as some decisions about splitting or merging of mentions with sub-entities ("Florida Supreme Court", "the State Council's Port Office"). Furthermore (3), we note that the additional named entity labels (Wikipedia links) are not complete, which we hope to change in future versions of CLEANCONLL. (4) Concerning tokenization and sentence segmentation, there still are some problems in the data, e.g. the problematic usage of hyphens ("-") inside tokens ("the Syrian-Lebanese peace tracks", "the India-South Africa test") which impedes proper labeling of the components ("Syrian", "Lebanese", "India", "South Africa").

In general, it might be a promising direction to allow for *multiple* or *nested* annotation possibilities. Both concerning label (e.g. there might be irresolv-

able ambiguities like "fellow Queenslander Daniel Herbert" referring to the city or sports team) as well as mention boundary decisions ("Thai Commerce Ministry").

Regarding general annotation quality, note that we only had two annotators – overlapping with the authors of this work. This might have affected labeling decisions or, in cases of doubt, pushed for correcting instead of keeping the label. However, as the authors worked extensively with the corpus throughout all of this work and had a very round view of their annotation guidelines, they favoured consistency instead of "conservatively" keeping and potentially overlooking label errors or edge cases. Providing the annotators with the source of the label suggestions during cross-checking instead of letting them choose completely "blind" was done intentionally to enable them to spot inconsistencies in the gold labels as well as mark cases for specific care which could then be incorporated in the succeeding rounds. However, appointing the same annotators for evaluation (annotation quality assessment and classification of model errors, see Sec. 3.2 and 4.2) is indeed not ideal but unfortunately could not be avoided due to time constraints and resources.

Concerning generalizability: We acknowledge that the initial round of leveraging entity linking annotations requires precisely such annotations and thus cannot be transferred to *every* NER dataset and other languages. Furthermore, the applied heuristics for deriving NER labels from the Wikidata taxonomy were constructed rather intuitively and would need updating in case of different, maybe more fine-grained NER labels (refer to Appendix A.2 for a discussion) and languages. However, we are confident that this is feasible, e.g. adding rules for "work of art", "medication", "physical unit". Furthermore, the cross-checking rounds are independent of the named entity linking labels and language of the dataset.

## Ethics Statement

This work does not raise many ethical problems in our opinion. The major concern is that CLEAN-CONLL still uses the same text base as the original: News articles that are over 20 years old. So both the language use and also the themes might be outdated and might also show signs of bias. Furthermore, we only worked on the English part of CoNLL-03.

---

[10]E.g. labeling airports as ORG and deities as MISC, deciding on LOC for "Vatican" and between MISC and ORG for "catholic (church)", labeling time zones ("GMT") as MISC but not labeling physical units etc.

## Acknowledgements

We thank the reviewers for their helpful comments. Susanna Rücker is supported by the German Federal Ministry of Economic Affairs and Climate Action (BMWK) as part of the project SEBIRA (KK5148003LB1). Alan Akbik is supported by the Deutsche Forschungsgemeinschaft (DFG, German Research Foundation) under Germany's Excellence Strategy "Science of Intelligence" (EXC 2002/1, project number 390523135) and the DFG Emmy Noether grant "Eidetic Representations of Natural Language" (project number 448414230).

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

# A   Appendix

## A.1   Mapping: Wikidata categories to NER

Through the Wikidata relations `instance_of` and `subclass_of`, for each Wikipedia URL in AIDA we get a collection of general categories like "national sports team" or "geographic region". From these collections, we derive the NER label (PER, ORG, LOC) with the following rules:

- ORG if collection includes one or more of ["organization", "political party", "political organization", "confederation", "sports club", "political party", "business", "public company", "type of organisation", "national sports team", "association football club", "sports team", "government organization"]

- LOC if collection includes one or more of ["state", "country", "city", "classification of human settlements", "human settlement", "geographic entity", "physical location", "U.S. state", "historical country", "island", "geographic region"]

- PER if collection includes one or more of ["human", "person", "Wikimedia human name disambiguation page"]

**Manual assignment for dummy MISC.** If either none or multiple of ORG, LOC or PER was assigned, we assign the MISC label instead, which for the time being serves as a dummy label marker for need of manual assignment. These were then manually checked and if necessary changed to ORG, LOC or PER.

## A.2   Analysis: Revisiting the Automatic NER Mapping

Having used automatic NER labeling through Wikipedia and Wikidata for initial corpus creation (recall Phase 1, Sec. 2.1), but then manually detecting and correcting labels through the cross-checking phases (Sec. 2.2), we are interested in the final picture in CLEANCoNLL, with respect to NER labels on the one hand and Wikipedia/Wikidata items on the other hand. This also gives insights into how reliable a purely automatic mapping approach could be when leveraging other NEL datasets for NER corpus correction or derivation. We conduct two analyses concerning a) automatic mapping correctness and b) alignment of NER type and Wikidata categories.

**Automatic Mapping Correctness.** The AIDA corpus uses 5,595 unique Wikipedia links. As we allowed our annotators to change or add new ones, our corpus version now includes 5,943 unique links. We look at how their current NER labels in

| Wikipedia–NER mapping in CLEANCONLL compared to | Automatic | | NER4NEL | |
|---|---|---|---|---|
| | abs | % | abs | % |
| Wiki label agrees | 5,101 | 85.8 | 5,278 | 88.8 |
| Wiki label differs | 842 | 14.2 | 438 | 7.4 |
| Dummy-MISC | 790 | 13.3 | | |
| Real difference | 52 | 0.9 | | |
| NA | 0 | 0.0 | 227 | 3.8 |
| # Wiki labels | 5,943 | | 5,943 | |

Table 5: Comparing the final Wikipedia–NER mapping in CLEANCONLL with the purely automatic one (Round A) and with the NER4NEL mapping by Tedeschi et al. (2021).

CLEANCONLL– after several cross-checking revision rounds – differ from the initial ones derived via the automatic approach in Phase 1 via Wikidata categories. To enable a full comparison, we rerun the automatic labeling method via Wikidata also for the 953 new Wikipedia links. The results of this comparison are in Table 5.

We note that 85.8% of the final labels agree with their automatically derived one from Phase 1, which overall shows that the automatic labeling process works well. The remaining 14.2% now have a different assigned NER label. This number seems alarmingly large at first. However, reall that we designed the rules for automatic NER extraction from Wikidata classes in favour of being more precise than comprehensive and used MISC as the initial dummy label when none (or multiple) of our heuristic rules applied to the item. This dummy MISC then led to the item being manually classified, so it is not surprising that the label now differs from MISC. In fact, not counting these dummy-MISC cases, *real* difference is only observed in 52 cases (0.9%) [11]. This shows, that the automatic NER mapping already produced high quality NER labels that nearly needed no revision.

If we were to purely use the automatic labeling approach through Wikipedia/Wikidata items, the heuristics could be further refined depending on the use case and particularly, one could add specific ones for the MISC label or any other additional label, considering the specific domain.

We add a comparison of our mapping to the one provided by Tedeschi et al. (2021) (NER4NEL) to Table 5, who also create NER labels for Wikipedia articles, but use the WordNet taxonomy instead of Wikidata. Since they use more fine-grained NER classes (18), we merge all of their classes other than PER, LOC and ORG to MISC for comparison. We note that our mapping agrees in 88.8%, but differs in 7.4%[12]. Looking through these cases of disagreement, there are cases where both NER decisions are understandable: "Metropolitan_Museum_of_Art" is labeled ORG in CLEANCONLL, but LOC in NER4NEL, newspapers like "The_Jakarta_Post" are labeled MEDIA in NER4NEL (which for the sake of comparison we merge to MISC) but ORG in CLEANCONLL. However, a good portion of the disagreements stem from definitely incorrect labels in NER4NEL, e.g. one sports club ("FC_Schalke_04") being labeled as LOC, others ("Sheffield_Eagles", "Gil_Vicente_F.C.") even as PER and "Supreme_Court_of_New_South_Wales" as TIME (so merged to MISC), where CLEANCONLL labeled ORG.

**NER labels and Wikidata categories.** Finally, we take a look at the final picture of NER labels and their respective most common Wikidata categories, i.e. the classes derived via the Wikidata instance_of and subclass_of relations, see Sec. 2.1. In Table 6 we look at the most frequent Wikidata classes that mentions with the respective NER label carry in CLEANCONLL (of course, only looking at those that *have* a Wikipedia label). We see, that the alignments intuitively make sense: For LOC we e.g. find "city", "state", "country" to be under the most frequent ones, for "ORG" we find "football club", "organization" and "business". Particularly interesting are the classes for the – now real – MISC label (e.g. "sports competition", "recurring event" or "(stock) exchange"), which possibly could be used to find additional rules for future automatic labeling methods. Reversely, for more specific domains, new NER labels could be constructed via frequently appearing Wikidata classes (e.g. "work of art", "event").

## A.3 Format of Dataset Distribution

Our resource enables deriving our dataset in CoNLL column format with BIO tagging scheme. See the excerpt in Figure 6 as an example: We pro-

---

[11]Many of these are cases in which we decided upon specific label usages *after* the heuristics and automatic labeling was already done, e.g. to label stock exchanges as MISC or universities and hospitals as ORG.

[12]The remaining 3.8% ("NA") are Wikipedia articles that are not present in the NER4NEL mapping, presumably because the article did not exist at the time or changed its name.

| NER label | Wikidata class | frequency |
|---|---|---|
| LOC | | |
| | city | 459 |
| | big city | 350 |
| | city or town | 338 |
| | urban area | 273 |
| | state | 213 |
| | human settlement | 209 |
| | political territorial entity | 209 |
| | country | 166 |
| | sovereign state | 157 |
| | capital city | 134 |
| ORG | | |
| | football club | 574 |
| | association football club | 527 |
| | sports team | 289 |
| | juridical person | 287 |
| | organization | 275 |
| | business | 256 |
| | association football team | 209 |
| | economic entity | 200 |
| | enterprise | 164 |
| | national sports team | 122 |
| PER | | |
| | human | 2,448 |
| | person | 2,448 |
| | omnivore | 2,448 |
| | natural person | 2,448 |
| | mammal | 2,448 |
| | Homo sapiens | 2,284 |
| MISC | | |
| | sports competition | 52 |
| | recurring event | 45 |
| | recurring sports event | 40 |
| | sports league | 35 |
| | tournament | 29 |
| | competition | 25 |

Table 6: Most frequent Wikidata classes (i.e. derived through `instance_of` and `sublass_of` relation) of mentions with the respective NER label.

vide the original POS tags from (Reiss et al., 2020) and our updated NER labels in the two variants (before and after reverting adjectival affiliations to MISC, see Sec. 2.3), as well as the Wikipedia links in the underscored version as they appear in the URL (e.g. Indonesia_national_football_team). See Appendix A.4 for a discussion of pros and cons of both label sets.

## A.4 Discussion: Dataset Variants on Adjectival Affiliations

As mentioned in Sec. 2.3, we distribute CLEAN-CoNLL with two sets of NER labels: One without and one with the reverts to MISC from Phase 3. We here discuss differences and (dis-)advantages of those two versions.

There are two main **advantages** in using the final variant (*with* MISC reverts):

1. Users of our resource might prefer the original notion of MISC to make results more comparable to the vast research and models trained on the original CoNLL-03.

2. A lot of the cases in question are actually cases with real ambiguity in meaning that makes it hard to decide on LOC or ORG and makes inconsistency in labels rather inherent. See for example the following sentences:

   (a) the [Chinese]LOC police
   (b) the [Chinese]ORG/LOC keeper
   (c) the [Chinese]LOC/ORG success in the skiing World Cup

   Here, the label depends on the understanding of the entity "Chinese" meaning (a) the country, (b) the national team *or* the nationality of the individual, or even (c), when there is not one specific team, but still some ORG-like semantics in several individuals representing the country. Using MISC (recall that the original CoNLL-03 and the Reiss version would have MISC in all of the three) erases the necessity of deciding on one meaning and therefore this specific source of inconsistency.

   Experiments comparing both variants of CLEANCoNLL show slightly better model performance on the final version *after* MISC reverts (see Sec. 4.1), which is a strong indicator for better label consistency.

However, there are some **downsides** to consider when using the final version with MISC reverts:

1. The ambiguity problem (the "Chinese" examples above) still remains in *non*-adjectival cases: In "it was China who dominated the games", the label still needs to be LOC or ORG, depending on the interpretation. Reusing MISC could actually make it harder for models to get a round understanding of the real LOC, ORG and MISC semantics and differentiation.

2. It is questionable to apply a purely syntactic criterion such as *adjectival use* in an otherwise more semantically influenced labeling task like NER. For illustration, consider the sentence "The area is patrolled by U.S., French and British planes." "French" and "British"

```
Text            POS     Wikipedia                               CleanCoNLL
                                                                before Phase 3    after Phase 3

Indonesian      JJ      B-Indonesia_national_football_team      B-ORG             B-MISC
keeper          NN      O                                       O                 O
Hendro          NNP     B-Hendro_Kartiko                        B-PER             B-PER
Kartiko         NNP     I-Hendro_Kartiko                        I-PER             I-PER
produced        VBD     O                                       O                 O
a               DT      O                                       O                 O
string          NN      O                                       O                 O
of              IN      O                                       O                 O
fine            JJ      O                                       O                 O
saves           VBZ     O                                       O                 O
to              TO      O                                       O                 O
prevent         VB      O                                       O                 O
the             DT      O                                       O                 O
Koreans         NNPS    B-South_Korea_national_football_team    B-ORG             B-MISC
increasing      VBG     O                                       O                 O
their           PRP$    O                                       O                 O
lead            NN      O                                       O                 O
.               .       O                                       O                 O
```

Figure 6: An excerpt from CLEANCONLL in column format, including one sentence with the original POS tags, Wikipedia labels and the two versions of updated NER annotations in BIO format.

are labeled MISC, whereas "U.S." gets LOC, even if the usage is quite the same. Befor reverting to MISC, all three of "U.S.", "French" and "British" are labeled LOC – which actually is more consistent.

3. If in a use case, NER is used as a method for collecting sentences or sentence parts that talk about a specific content, say, specific (national) sports teams or searching for place or nationality related elements in documents, it would be harder to catch the MISC labeled entities as well. However, thanks to our Wikipedia labels, this problem might be mitigatable.

We leave it up to the researchers to choose which variant of CLEANCONLL is a better fit for their use case.

## A.5 Qualitative Examples for Errors in CoNLL-03

We present examples for errors and inconsistencies found in the previous corpus versions (CoNLL-03 and the version by Reiss et al. (2020)) in comparison to their annotation in CLEANCONLL – both *before* and *after* the reverts to MISC in Phase 3. See Table 7 for the listed examples with their full NER labels. We find:

- **Wrong labels.** Some labels are outright incorrect, like in (a), where a person's first

name "Florence" is confused with the city (labeled LOC, corrected in Reiss), or (b), where "Green" is spuriously labeled as PER, but in fact is just the color or (c), where "Mind Games" is labeled as PER (still in Reiss), when in fact it is the name of a racehorse (thus gets MISC from our annotators). In (d), "William Hill" is labeled as PER in CoNLL-03 and Reiss, but refers to a gambling company with the same name (ORG).

- **Missing mentions.** In (e), one of three city names ("Manaus") is missing the LOC label in CoNLL-03 and Reiss. Similarly in (f), where "Thome" was missed as a person's name in both, or (g), where we decided to add a label to "Fascist".

- **Boundary problems.** In (h), an incorrect sentence boundary leads to missing part of the mention ("Hamlet Cup"). Interestingly, Reiss correct the sentence, but still are missing the full mention. In (i), missing spacing around the dash is corrupting the labels: "GOETCHL" is a person, "ALPINE" should be labeled, though the type is arguable. In (j), we decided to extend the mention to "Arthur Yates and Co ltd".

- **Label inconsistencies.** In both (k) and (l), "Australia" refers to national sports teams. Whereas the CoNLL-03 guidelines specifi-

cally state to use LOC in this case (so this is not really an annotation *error* per se), we disambiguate between references to the country or sport team and use ORG accordingly. The Reiss labeling is inconsistent: LOC in (k) and ORG in (l) – introducing inconsistency, a very frequent source of noise in their dataset. Similarly, both (m) and (n) are structured "[Team1] at [Team2]". While "Team2" is labeled consequently as LOC in CoNLL-03 (an understandable ambiguity, since "Team1" is both playing against "Team2" and *at "Team2"'s home site*), Reiss introduce inconsistency by changing the label to ORG in some cases (n), but keeping LOC in others (m).

## A.6 Updated label guidelines for CLEANCoNLL

During our relabeling phases, we updated some labeling guidelines compared to the label usage in original CoNLL-03, Reiss et al. (2020) and Wang et al. (2019). All of these were done to ensure better consistency throughout the corpus. More concrete:

- National sports teams are frequently referred to by country name ("Japan began the defence of their Asian Cup title"). The original annotation guidelines from CoNLL-03 state that in this case the country name should still be labeled as LOC nonetheless (Chinchor et al., 1999). We change this and use ORG in these cases because we deem it more consistent to other (not national) sports team usages ("during Milan's 2-1 defeat" or "the Colts who played their last home game")[13].

- There are several other cases where we observed label variation and decided on a fixed guideline to ensure consistency. Whereas there is label variation between LOC and ORG for e.g. airports, hospitals, schools, universities, we label them consistently as ORG. For sports leagues, sport events and stock exchanges we use MISC consistently.

- Several mentions include a compound with an e.g. location and adjective or adverb ("the Washington-based Consumer Project

on Technology", "German-born U.S. biologist Max Delbruck", "the current NATO-led peace force"). Whereas these cases mostly have MISC before (sometimes they are unlabeled), we decided to consequently use the label that the entity part ("Washington", "German", "NATO") would get on its own.

- Oftentimes mentions could be labeled either as one longer or as consisting of several shorter mentions ("the 11th Circuit U.S. Court of Appeals in Atlanta", "the 1994 Lillehammer Winter Olympics"). We split them up, if the parts independently still form a frequent mention on their own ("Atlanta", "Lillehammer"), but merge if the parts are infrequent or hard to interpret ("11th Circuit U.S. Court of Appeals").

- All of CoNLL-03, the version by Reiss et al. (2020), and the one by Wang et al. (2019), use the MISC label for adjectival affiliations ("Uzbek striker Igor Shkvyrin", "the Italian club"). Our initial CLEANCoNLL labels use ORG or LOC, according to the referred entity. However, in Phase 3 (recall Sec. 2.3) we revert these cases to MISC, so in final CLEANCoNLL, the label usage is conform to the predecessor versions. Note that we do distribute the labels *before* the reverts as well.

## A.7 Details on Entity Classification Model

In the second round of cross-checking (Sec. 2.2.2) we again trained 20 models in a cross-checking fashion for spotting inconsistencies. However, this time we simplified the task and let the models classify already given fixed mention spans into NER types, i.e. to perform only Named Entity *Classification* and not Detection. Instead of sequence tagging like in the previous cross-checking round, we used a span tagging model that represents the given mentions as concatenation of the first and last token's contextualized embeddings. Putting aside the task of mention detection allows the model to really focus on the label semantics. In contrast, recall that in the third round of cross-checking, we instead use a model that *only* focuses on mention detection and ignores NER clssification.

---

[13]Reiss et al. (2020) mostly do so as well, though they do not state this in their paper – however not consistently, see examples in Appendix A.5. Wang et al. (2019) do not change these cases.

| | Sentence (part) | Labels in CoNLL-03 (Tjong Kim Sang and De Meulder, 2003) | Labels in Reiss (Reiss et al., 2020) | Labels in CLEAN-CONLL before Phase 3 | Labels in CLEAN-CONLL after Phase 3 |
|---|---|---|---|---|---|
| (a) | 11. Florence Masnada (France) 133 | [Florence]LOC [Masnada]PER [France]LOC | [Florence Masnada]PER [France]LOC | [Florence Masnada]PER [France]LOC | [Florence Masnada]PER [France]LOC |
| (b) | Green and black face paint completed his disguise. | [Green]PER | [Green]PER | | |
| (c) | Favourite: Mind Games (7-4) finished 4th | [Mind Games]PER | [Mind Games]PER | [Mind Games]MISC | [Mind Games]MISC |
| (d) | UK bookmakers William Hill said on Friday they have lengthened the odds of a Conservative victory | [UK]LOC [William Hill]PER [Conservative]MISC | [UK]LOC [William Hill]PER [Conservative]MISC | [UK]LOC [William Hill]ORG [Conservative]ORG | [UK]LOC [William Hill]ORG [Conservative]MISC |
| (e) | the cities of Manaus, Sao Paulo and Rio de Janeiro. | [Sao Paulo]LOC [Rio de Janeiro]LOC | [Sao Paulo]LOC [Rio de Janeiro]LOC | [Manaus]LOC [Sao Paulo]LOC [Rio de Janeiro]LOC | [Manaus]LOC [Sao Paulo]LOC [Rio de Janeiro]LOC |
| (f) | With the score tied 1-1 in the ninth, Thome hit a 2-2 pitch | | | [Thome]PER | [Thome]PER |
| (g) | the granddaughter of Italy's Fascist dictator Benito Mussolini | [Italy]LOC [Benito Mussolini]PER | [Italy]LOC [Benito Mussolini]PER | [Italy]LOC [Fascist]MISC [Benito Mussolini]PER | [Italy]LOC [Fascist]MISC [Benito Mussolini]PER |
| (h) | (Results at the Hamlet) Cup tennis tournament[1] | [Cup]MISC | [Cup]MISC | [Hamlet Cup]MISC | [Hamlet Cup]MISC |
| (i) | ALPINE SKIING - GOETCHL WINS WORLD CUP DOWNHILL.[2] | [WORLD CUP]MISC | [WORLD CUP]MISC | [ALPINE]LOC [GOETCHL]PER [WORLD CUP]MISC | [ALPINE]LOC [GOETCHL]PER [WORLD CUP]MISC |
| (j) | NOTE: Arthur Yates and Co ltd is a garden products group. | [Arthur Yates and Co]ORG | [Arthur Yates and Co]ORG | [Arthur Yates and Co ltd]ORG | [Arthur Yates and Co ltd]ORG |
| (k) | the man who kicked Australia to defeat | [Australia]LOC | [Australia]LOC | [Australia]ORG | [Australia]ORG |
| (l) | He will not be considered for a test match against Australia starting on October 10 | [Australia]LOC | [Australia]ORG | [Australia]ORG | [Australia]ORG |
| m) | LA CLIPPERS AT NEW YORK | [LA CLIPPERS]ORG [NEW YORK]LOC | [LA CLIPPERS]ORG [NEW YORK]LOC | [LA CLIPPERS]ORG [NEW YORK]ORG | [LA CLIPPERS]ORG [NEW YORK]ORG |
| n) | CALGARY AT BOSTON | [CALGARY]ORG [BOSTON]LOC | [CALGARY]ORG [BOSTON]ORG | [CALGARY]ORG [BOSTON]ORG | [CALGARY]ORG [BOSTON]ORG |

Table 7: Example sentences or sentence parts and their respective labels in CoNLL-03, Reiss and our two CLEANCONLLversions. [1]In CoNLL-03 the sentence is incorrectly split before *Cup*. While in Reiss this is corrected, the label still is only placed on *Cup*, not on *Hamlet Cup*. [2]In CoNLL-03 and Reiss, the spacing around the dash is missing, making "SKIING-GOETCHL" one token and therefore making the correct labeling impossible. In CLEANCONLLwe added the spaces and missing mention.

| | CoNLL-03 | | Wang | | Reiss | | CLEANCONLL[1] | | CLEANCONLL | | AIDA |
|---|---|---|---|---|---|---|---|---|---|---|---|
| | abs | % | abs | % | abs | % | abs | % | abs | % | abs |
| LOC | 10,645 | 30.3 | 10,623 | 30.2 | 10,103 | 28.9 | 12,089 | 34.3 | 9,399 | 26.7 | |
| ORG | 9,323 | 26.6 | 9,377 | 26.7 | 9,922 | 28.4 | 10,701 | 30.4 | 10,492 | 29.8 | |
| PER | 10,059 | 28.7 | 10,060 | 28.6 | 9,983 | 28.6 | 9,947 | 28.2 | 9,947 | 28.2 | |
| MISC | 5,062 | 14.4 | 5,083 | 14.5 | 4,933 | 14.1 | 2,520 | 7.1 | 5,419 | 15.4 | |
| # entities | 35,089 | | 35,143 | | 34,941 | | 35,257 | | 35,257 | | 27,814 |
| # sentences | 20,744 | | 20,744 | | 20,617 | | 20,617 | | 20,617 | | 20,584 |

Table 8: Statistics of both label set variants from CLEANCONLL, in comparison to previous versions.
[1]CLEANCONLL labels **before** reverts in Phase 3.

**Reiss version**
vs. CoNLL-03

| | abs | % |
|---|---|---|
| Label unchanged | 33,340 | 95.0 |
| Label updated | 743 | 2.1 |
| New mention | 207 | 0.6 |
| Unknown | 651 | 1.9 |
| # entities | 34,941 | |
| # entities orig. | 35,089 | |
| # updated sent. | 969 | 4.7 |

Table 9: NER label updates, comparing mentions in Reiss version with CoNLL-03.

**CLEANCONLL**
vs. AIDA

| | abs | % |
|---|---|---|
| URL unchanged | 32,699 | 92.7 |
| URL updated | 322 | 0.9 |
| URL added | 1,955 | 5.5 |
| Unknown | 281 | 0.8 |
| # entities | 35,257 | |
| # entities orig. | 27,814 | |
| # updated sent. | 2,077 | 10.1 |

Table 10: Comparing Wikipedia labels in CLEAN-CONLL to the original ones from AIDA.

*Unknown*: We map and compare the labels using the sentence text. Due to updated sentence boundaries from the original CoNLL-03, in some cases we cannot find a direct equivalent, so we exclude the respective mentions here.
*New mention*: Includes both completely new as well as newly bounded mentions.

## A.8 Additional Dataset Statistics

### A.8.1 Dataset Statistics – Full comparison

In Table 8 (an extension to Table 2) we present a full comparison of label distribution and general dataset statistics, comparing CLEANCONLL to previous versions, as well as AIDA. We include both versions of CLEANCONLL NER labels, so the labels *before* and *after* the reverts to MISC in Phase 3 (recall Sec. 2.3). Most striking is the decrease in MISC usage before the reverts, and the subsequent increase of the LOC and ORG, coming from labeling adjectival affiliation as either ORG or LOC, according to their original label that we derived through the entity linking approach. In the final version, the distribution is more similar to the previous versions.

### A.8.2 Label Updates: Reiss and Wikipedia

In Table 9 we present the amount of label updates that the version by Reiss et al. (2020) introduced in comparison to CoNLL-03 (see the CLEANCONLL updates in Table 3 for comparison).

Table 10 shows the amount of updates from CLEANCONLL concerning the Wikipedia URLs: 92.7% of labels are unchanged in comparison to AIDA, 0.9% were updated from our annotators, and 5.5% of mentions (1,955) got a newly added Wikipedia label.

### A.8.3 Label Updates: Before and after MISC reverts

In Tables 11 and 12 we contrast the label updates introduced by CLEANCONLL, comparing it to the previous versions. We present the state of NER labels *before* (Table 11) and *after* (Table 12) the final Phase 3 reverts from Sec. 2.3. Note that the amount of label updates before applying our special handling of adjectival affiliations (and keeping ORG and LOC instead) is accordingly much higher (14.9% label updates, 20.2% updated sentences) compared to the final 7.0% label updates (10.3% sentences). Refer to Appendix A.4 for a discussion of which label set to use.

| **CLEANCONLL** (before Phase 3) | | | | | | |
|---|---|---|---|---|---|---|
| vs. | CoNLL-03 | | Wang | | Reiss | |
| | abs | % | abs | % | abs | % |
| Label unchanged | 29,348 | 83.6 | 29,452 | 83.8 | 30,671 | 87.8 |
| Label updated | 4,595 | 13.1 | 4,557 | 13.0 | 4,023 | 11.5 |
| New mention | 624 | 1.8 | 558 | 1.6 | 533 | 1.5 |
| Unknown | 690 | 2.0 | 690 | 2.0 | 30 | 0.1 |
| # entities | 35,257 | | 35,257 | | 35,257 | |
| # entities orig. | 35,089 | | 35,143 | | 34,941 | |
| # updated sentences | 4,169 | 20.2 | 4,095 | 19.9 | 3,550 | 17.2 |

Table 11: NER label updates, comparing CLEAN-
CONLL **before reverts in Phase 3** to the previous
versions.

| **CLEANCONLL** (after Phase 3) | | | | | | |
|---|---|---|---|---|---|---|
| vs. | CoNLL-03 | | Wang | | Reiss | |
| | abs | % | abs | % | abs | % |
| Label unchanged | 32,110 | 91.5 | 32,224 | 91.7 | 33,488 | 95.8 |
| Label updated | 1,833 | 5.2 | 1,785 | 5.1 | 1,206 | 3.5 |
| New mention | 624 | 1.8 | 558 | 1.6 | 533 | 1.5 |
| Unknown | 690 | 2.0 | 690 | 2.0 | 30 | 0.1 |
| # entities | 35,257 | | 35,257 | | 35,257 | |
| # entities orig. | 35,089 | | 35,143 | | 34,941 | |
| # updated sentences | 2,115 | 10.3 | 2,027 | 9.8 | 1,398 | 6.8 |

Table 12: NER label updates, comparing CLEAN-
CONLL **after reverts in Phase 3** to the previous
versions.