# OpenReview forum: "CleanCoNLL: A Nearly Noise-Free Named Entity Recognition Dataset"
_EMNLP/2023/Conference — EMNLP 2023 Main_

### Official Review · Reviewer_NcfC · 2023-08-02

**Soundness:** 4

**Excitement:**

4: Strong: This paper deepens the understanding of some phenomenon or lowers the barriers to an existing research direction.

**Paper Topic And Main Contributions:**

This paper describes an effort for improving the quality of the annotation of the CoNLL corpus. The presented methodology consists in several two-step correction rounds. In the first step the authors leverage automatic methods for highlighting potential errors in the original corpus. In the second step authors mobilized annotators for judging the cases raised in the first step.
The authors describe three methods for highlighting potential errors:
- comparison of CoNLL NE labels with AIDA EL labels
- cross-checking by training NER tools and label classifiers
- training a NER model specifically for adjectival affiliation

The authors also evaluated the quality improvement in two ways:
- manual reassessment of the corrected annotation
- comparison of SOTA methods trained on different versions of the dataset

**Questions For The Authors:**

- general : in this work, were annotators the same persons that conceived the whole correction framework? If so, don't you fear it would introduce a bias (pushing for corrections)?
- general : most examples given in this paper concerns sports, and especially one phenomenon (LOC/ORG confusion for teams). Does this actually represent most of the improvements in CleanCoNLL, or is it just an example? Could you clarify that?
- line 255 : source of the Wikipedia label provided to annotators (AIDA? prediction?)
- line 255 and Figure 3 : the annotators did know which label was the original and which was predicted. Don't you think this might introduce a bias in the correction?
- line 348 : this line states that there have been boundary corrections, however section 2 implies only label modifications and entity additions, could you clarify at which phases and how boundaries were corrected?
- Table 3 : are there significant differences in the modification rate between train and test splits? Line 047 implies that this would be possible. If so how would that affect experiments of section 4.2?
- line 442 : please explain why you chose to examine errors of FLERT whereas Biaffine seems to yield better results?

**Reasons To Accept:**

1. Shares a very useful resource

2. Convincing methodology and quality assessment

3. Paper is clearly written, describes thoroughly the motivation, the methodology, the results and the limitations

**Reasons To Reject:**

None that I can think of.

As the authors state in the Section "Limitations", the methodology they used is not apply on other corpora, especially domain-specific corpora. However I'd argue that this paper might inspire future dataset creators for quality checking methods.

**Reproducibility:**

4: Could mostly reproduce the results, but there may be some variation because of sample variance or minor variations in their interpretation of the protocol or method.

**Reviewer Confidence:**

3: Pretty sure, but there's a chance I missed something. Although I have a good feel for this area in general, I did not carefully check the paper's details, e.g., the math, experimental design, or novelty.

**Typos Grammar Style And Presentation Improvements:**

- line 047 : of all sentences -> in all sentences
- Figure 3 : please change background color, green and blue texts on dark red are difficult to read for some people
- line 270 : Some -> some

---

> ### Author Rebuttal · Authors · 2023-08-25
>
> Thank you very much for your elaborate review with a lot of helpful comments and fair questions! We are very happy that you find the paper and methodology convincing and clear, as well as deem our resource helpful and point out our assessments of annotation quality!
>
> As for your specific questions:
> 1. Yes, the annotators do overlap with the authors. You are right that this could influence annotations and pushing for modifications. However, this could be seen also as a good thing because having worked extensively with the dataset and quality assessment, the annotators were very aware of inconsistencies and presumably had a more round view of annotation guidelines which pushes for consistency rather than “conservatively” keeping the labels as is. However, appointing the same annotators for manual evaluation (deciding between annotation and model errors) is indeed not ideal but unfortunately could not be avoided due to time constraints and resources. We will state this more clearly in the final version.
> 2. The observation that most of the examples concern sports is interesting! Many articles in CoNLL (especially the test set) are sports themed so these do make up for a considerable portion of changes, but by no means all, as is already shown by Table 1 and Table 7. For example, hospitals/schools show some similar ambiguous handling, as do non-sports governmental organizations such as armed forces. We will make sure to have more diverse examples in the final version!
> 3. Yes, the annotators were provided with information as to where a label suggestion comes from, e.g. gold annotation, model prediction or automatically derived via Wikipedia/Wikidata heuristics.
> 4. Refer to 3. and 1. above. You are right, this could well have been done “blindly” with some benefits. However, stating the label source more clearly was also helpful: It made inconsistencies in the gold labels more strikingly clear to annotators (as well as could have shown general problems with our automatic NER labels!), and enabled them to mark cases for specific care (see line 265), which could then be incorporated in the succeeding rounds.
> 5. As stated in Section 2.2.2, in the third round of cross-checking (see line 291 “3. Only entity detection.”), we specifically mined for boundary problems and corrected them.
> 6. This is an interesting idea, thanks! Unfortunately we have not checked for differences of correction rates in the splits but will do so and add them to Table 3.
> 7. We used FLERT since the Biaffine approach only achieved the best numbers on CleanCoNLL. On the other CoNLL variants we evaluate, FLERT slightly outperformed or was on par with the Biaffine approach. Using Biaffine would therefore have slightly biased the evaluation to favor CleanCoNLL, which we wanted to avoid for fairness.
>
> We hope this helps resolve some of your questions! Furthermore, thank you for marking typos and the helpful hint regarding a more inclusive color usage in Figure 3!

---

### Official Review · Reviewer_xzrG · 2023-08-04

**Soundness:** 4

**Excitement:**

3: Ambivalent: It has merits (e.g., it reports state-of-the-art results, the idea is nice), but there are key weaknesses (e.g., it describes incremental work), and it can significantly benefit from another round of revision. However, I won't object to accepting it if my co-reviewers champion it.

**Missing References:**

There is another relabeling effort on CoNLL English: https://ojs.aaai.org/index.php/AAAI/article/view/6276 (Rethinking Generalization of Neural Models: A Named Entity Recognition Case Study). It will be nice to add this dataset into comparison as well.

**Paper Topic And Main Contributions:**

This paper presents a relabeling effort for fixing the annotation error from the CoNLL-2003 English dataset. They propose a pipeline that are composed of entity linking to AIDA, cross-checking and fixing adjectival affiliations, which greatly reduce the labeling noise comparing the the original version of the dataset.

**Reasons To Accept:**

- the paper contributes a new version of relabeled CoNLL NER dataset with fewer annotation errors
- the paepr conducts comprehensive analysis on the annotation evaluation, result evaluation, etc. readers can understand the statastics of the dataset much better through the thwe analysis
- it's interesting to see the techniques / tasks of entity linking and cross checking to be applied to fix annotation errors

**Reasons To Reject:**

I didn't see major weaknesses regarding the technical aspects. My main doubt or question though, is do we need another version of CoNLL-2003 English. NER people have been fully aware of the annotation error issues from the CoNLL dataset, as it's also mentioned in the paper. From the modeling anfd evaluation perspetive, fixing these errors also have limited ability to distinguish the model performance under the context of CoNLL-2003 English.
Personally, I would be more excited to see the labeling efforts and techniques being applied to datasets that are larger or in other langugages.

**Reproducibility:**

4: Could mostly reproduce the results, but there may be some variation because of sample variance or minor variations in their interpretation of the protocol or method.

**Reviewer Confidence:**

4: Quite sure. I tried to check the important points carefully. It's unlikely, though conceivable, that I missed something that should affect my ratings.

**Typos Grammar Style And Presentation Improvements:**

it's always a good idea to mention in the abstract that which language the paper is working on (e.g., #BenderRule). For this paper, I would be clear that it's only about CoNLL 2003 English dataset.

---

> ### Author Rebuttal · Authors · 2023-08-25
>
> Thank you very much for your helpful comments, the pointer to the paper to consider, as well as for your fair criticism concerning the limitations/transferability of the approach!
>
> You are right, we only consider the English version for now and will follow your suggestion to state this clearly in the abstract of the paper. This follows our initial relabeling round through the Named Entity Linking annotations that are exclusively available for the English CoNLL03 dataset (AIDA). However, though obviously not done in the present work, we do think that the proposed approach of a) leveraging NEL and Wikipedia/Wikidata entities for deriving automatic NER labels, and b) checking consistency via iterative cross-checking is generally applicable for other datasets and languages. The former obviously requires EL annotations while the latter is completely independent and in theory could be seen as a default consistency check for any dataset. We do hope to apply the approach to more datasets and if possible other languages in future work!
>
> Regarding novelty and usefulness, our model evaluation and inspection of model predictions do point to gains in performance and a more fair comparison when using our corrected dataset in training and evaluation compared to the original as well as other corrected versions of the dataset. In particular (see Section 4.2), evaluations on earlier CoNLL-03 versions had a large share of correct predictions falsely counted as errors due to annotation noise, making it difficult to analyze the remaining errors made by state-of-the-art systems. With CleanCoNLL, this noise is greatly reduced, allowing us to analyze the remaining errors, and indicating that the theoretical upper bound even on high resource, coarse-grained NER is not yet reached.

---

### Official Review · Reviewer_L4cT · 2023-08-04

**Soundness:** 4

**Excitement:**

4: Strong: This paper deepens the understanding of some phenomenon or lowers the barriers to an existing research direction.

**Paper Topic And Main Contributions:**

This paper presents a new version of the well-known CoNLL-03 corpus for NER. The new version, CleanCoNLL,
have undergone a comprehensive manual relabeling assisted by automatic consistency checking used during the manual re-annotation process to flag inconsistencies. The authors claim that that 7% of all labels in CoNLL-03 were corrected. The corrections were carried out on all three splits, and the dataset will be released and made available following the same copyright as the original data.

CleanCoNLL also contains more consistent NER annotations and integrated a layer of entity linking annotations. These latter are leveraged from Wikipedia and Wikidata. The authors also carried out three iterative rounds of cross-checking to identify potential inconsistencies. These were subsequently manually examined and corrected if necessary. The authors have also developed and added special handling for adjectival affiliation entity names.

The authors give an extensive experimental evaluation of the dataset compared to its original version, and using a set of stat-of-the-art (sota) approaches (Flair model, FLERT model, Biaffine tagging approach, and the ACE model (Wang et al., 2021)). They show that using the new clean dataset, sota approaches reach higher F1 scores with less misclassifications due to annotation errors. These types of errors dropped from 47% to 6% which a considerable drop.

The paper is well written and clear. This is a great contribution which I believe will help develop sota NER models for years to come.


**Reasons To Accept:**

- New corrected version of the well-known CoNLL-03 dataset.
- Corrections and additions made to the inconsistencies present in the annotations
- The corrections improve models trained on the data.
- The data will be made freely available.

**Reasons To Reject:**

- I cannot see any.

**Reproducibility:**

4: Could mostly reproduce the results, but there may be some variation because of sample variance or minor variations in their interpretation of the protocol or method.

**Reviewer Confidence:**

3: Pretty sure, but there's a chance I missed something. Although I have a good feel for this area in general, I did not carefully check the paper's details, e.g., the math, experimental design, or novelty.

---

> ### Author Rebuttal · Authors · 2023-08-25
>
> Thank you so much for your comments, we are very happy that you consider our paper and resource helpful for the community and appreciate your time in writing the review! We hope the dataset can indeed help development and fair comparison of future NER architectures.

---

### Meta-Review · Area_Chair_pm2x · 2023-09-12

**Recommendation:** 5

**Metareview:**

The paper presents a substantial contribution in the form of a corrected and enhanced version of the CoNLL-03 corpus, which helps improving reliability and accuracy of Named Entity Recognition (NER) evaluation. The main contribution of this work, apart from the valuable new dataset, is the interestingly novel methodology employed for the automatic correction and cleaning of the corpus. Although it’s transferability to other languages still has to be proven experimentally, it appears promising.

**Pros:**

- The paper is well written and clear

- The methodology is convincing and sound

- Produced a new corrected version of the well-known CoNLL-03 dataset, a very useful resource for the community

- gives detailed thorough statistics of the dataset, to assess its quality

- creative / innovative applications of known techniques to the task of annotation correction and cleaning

- the methodology seems portable to other languages


The work has no important weakness. It is interesting, mature, and the dataset it presents is likely to have a long-term impact on the field.

---

### Decision · Program_Chairs · 2023-10-07

**Decision:**

Accept-Main

**Comment:**

The paper presents a substantial contribution in the form of a corrected and enhanced version of the CoNLL-03 corpus, which helps improving reliability and accuracy of Named Entity Recognition (NER) evaluation. The main contribution of this work, apart from the valuable new dataset, is the interestingly novel methodology employed for the automatic correction and cleaning of the corpus. Although it’s transferability to other languages still has to be proven experimentally, it appears promising.

**Pros:**

- The paper is well written and clear

- The methodology is convincing and sound

- Produced a new corrected version of the well-known CoNLL-03 dataset, a very useful resource for the community

- gives detailed thorough statistics of the dataset, to assess its quality

- creative / innovative applications of known techniques to the task of annotation correction and cleaning

- the methodology seems portable to other languages


The work has no important weakness. It is interesting, mature, and the dataset it presents is likely to have a long-term impact on the field.